# Analysis of the Impact of Oral Health on Adolescent Quality of Life Using Standard Statistical Methods and Artificial Intelligence Algorithms

**DOI:** 10.3390/children8121156

**Published:** 2021-12-08

**Authors:** Milica Gajic, Jovan Vojinovic, Katarina Kalevski, Maja Pavlovic, Veljko Kolak, Branislava Vukovic, Rasa Mladenovic, Ema Aleksic

**Affiliations:** 1Faculty of Stomatology Pancevo, University Business Academy in Novi Sad, 26000 Pancevo, Serbia; milicaarsenijevic@gmail.com (M.G.); jovan.vojinovic@sfp.rs (J.V.); katarina.kalevski@sfp.rs (K.K.); maja.pavlovic@sfp.rs (M.P.); veljko.kolak@sfp.rs (V.K.); branislava.vukovic@sfp.rs (B.V.); ema.aleksic@sfp.rs (E.A.); 2Faculty of Medical Sciences, Department of Dentistry, University of Kragujevac, 34000 Kragujevac, Serbia

**Keywords:** adolescents, quality of life, oral health, artificial intelligence

## Abstract

The aim of this study was to determine the impact of oral health on adolescent quality of life and to compare the results obtained using standard statistical methods and artificial intelligence algorithms. In order to measure the impact of oral health on adolescent quality of life, a validated Serbian version of the Oral Impacts on Daily Performance (OIDP) scale was used. The total sample comprised 374 respondents. The obtained results were processed using standard statistical methods and machine learning, i.e., artificial intelligence algorithms—singular value decomposition. OIDP score was dichotomized into two categories depending on whether the respondents had or did not have oral or teeth problems affecting their life quality. Human intuition and machine algorithms came to the same conclusion on how the respondents should be divided. As such, method quality and the need to perform analyses of this type in dentistry studies were demonstrated. Using artificial intelligence algorithms, the respondents can be clustered into characteristic groups that allow the discovery of details not possible with the intuitive division of respondents by gender.

## 1. Introduction

Understanding and assessing the behavior of children and adolescents are complex endeavors because they are an unstable group, still developing skills and functions. On entering puberty, emotional factors, such as aesthetic values, increase in importance, while cognitive factors, such as acquired knowledge and behavior, weaken. Problems related to oral health may lead to adolescents feeling inferior, and lead to lower self-confidence and difficulties in performing daily activities. Poor oral health may be particularly problematic when enjoying food, speaking, sleeping, laughing, cleaning teeth, completing school duties, and creating friendly and emotional relationships [1].

Oral health is an important component of everyday functioning and well-being. Oral health can be measured by two essentially different approaches: through clinical parameters determined by dentist examination, or subjectively, where the individual independently assesses their own oral health. Subjective evaluation of oral health is known to affect the impact of oral health on quality of life. A large number of methods have been developed to explore the link between oral health and quality of life [2]. These scales have generally been designed and tested in the adult population. Among scales which measure the impact of oral health on the daily activities of young people, with reliability and validity in various cultural environments, the Oral Impacts on Daily Performance (OIDP) scale stands out [3]. The Serbian version of OIDP questionnaire was introduced in 2012 [4].

Through the use of questionnaires to measure the impact of oral health on daily activities (OIDP), a significant difference has been observed between objective and subjective assessments of the need for dental care [5]. Quality of life as a function of oral health measures subjective indicators based on an individual’s views on their own oral status and its impact on different aspects of their life. Oral symptoms, functional limitations, and social and emotional well-being are related and can be used to measure quality of life [6].

In order to determine the real needs of adolescents in terms of oral health and to set future goals of preventive and therapeutic programs, numerous public health and epidemiological studies have been conducted. As such, there are large amounts of data on the subject. Modern technology has devised ways to efficiently store data so as to occupy as little space as possible but preserve the main information. The continued advances made in the production of multimedia data implies constant development and further optimization of various forms of compression. An example of this is the compression of a digital image. A digital image is usually presented by a rectangular matrix of image elements—pixels, where each pixel is defined by a certain number of components presented by a corresponding number of bits (from 0 to 256). The use of artificial intelligence in contemporary dentistry clearly shows its significance in dental imaging, oral diagnosis and treatment, dental designs and dental research. Machine learning is an emerging field of artificial intelligence research and practice in which computer agents are employed to improve perception, cognition, and action based on their ability to “learn”, for example through use of big data techniques. Its application within dentistry is designed to promote personalized and precision patient care, with enhancement of diagnosis and treatment planning [7].

The aim of this study was to measure the impact of oral health on adolescent quality of life and to compare the results obtained by using standard statistical methods (*t* test—significance test) and artificial intelligence algorithms.

## 2. Materials and Methods

### 2.1. Respondents and Research Protocol

This study included 1st -and 2nd-year students from Belgrade Secondary Dental School and their peers who attended the Faculty of Stomatology Pancevo in the capacity of patients. The total sample comprised 374 respondents (128 respondents of male gender and 246 respondents of female gender). A random selection method was used to choose 249 respondents from the Secondary Dental School. The number of respondents who attended the Faculty of Stomatology and took part in this study was 125. Study methods and aims were explained to the respondents. All respondents and their parents agreed to participate in this anonymous study.

Implementation of this study was approved by the Ethics Committee of the Faculty of Stomatology Pancevo (No. 795/2-2016 from 26 May 2016).

### 2.2. The Impact of Oral Health on Adolescent Quality of Life

In order to measure the impact of oral health on adolescent quality of life, a validated Serbian version of the Oral Impacts on Daily Performance (OIDP) scale was used [5]. The scale measures how often in the last six months the respondents have had problems performing daily life activities due to oral and dental problems. The frequency of difficulties is measured on a five-point scale, where 1—never or less than once a month; 2—once to twice a month; 3—once to twice a week; 4—three to four times a week; 5—almost every day. The results of this questionnaire can provide insight into whether and how often adolescents have had problems during eating, speaking, cleaning teeth, sleeping, laughing, establishing emotional and social contacts, and completing school duties. The lowest score is 8 and indicates that the respondent does not have any oral health-related problems which affect performing daily life activities. In contrast, the maximum score is 40 and indicates the existence of a large number of oral health-related problems which negatively affect quality of daily life.

### 2.3. Artificial Intelligence Algorithms

A special program in the Python programming language [8] was used to parse data, create a digital database, and process data using standard statistical methods and SVD. The singular value decomposition (SVD) algorithm is based on the principles of reducing dimensionality by decomposition into singular values [9,10] with the aim of increasing the degree of image compression [11]. SVD uses the simple strategy of approximation of the original matrix that represents the digital image [12,13], using smaller matrices. The selection of a rank value for the matrix which is used after applying SVD to represent the original image matrix is a compromise between achieving the definite desired degree of compression and acceptable quality of the digital image. Matrix A can be decomposed into 3 matrices. (A = U × Σ × V).

SVD represents a relatively simple strategy of optimal approximation of the matrix by smaller matrices. Since the elements of matrix Σ (singular values) are sorted into descending order, it is possible to keep the first k biggest values and set others to zero. The product of obtained matrices in this case is a new matrix Ak, of k rank, which is a good approximation of matrix A [14,15]. The main diagonal (Σ) has elements which are sorted into descending order, and the first element is the most important factor. If we include only the most important factor and assign zero to other elements, we obtain a new matrix Σ′ (Sigma importance 1). By further calculation, we obtain a new A′ matrix, which has great similarity to the original matrix A (A′ = U × Σ′ × V).

Practical implementation of SVD allows the user to achieve the desired degree of compression, with acceptable calculation complexity. Using SVD, “help” can be obtained from machine learning in numerous analyses. It will give us guidelines on the way how we should cluster respondents.

Matrices that will be described in the results of our study will have 374 rows, which represent the number of respondents, and the number of columns will be 8 as there are that many questions in the questionnaire to which the respondents gave their answers [16]. On the basis of input data, and using the factorization method [17], one matrix (A) will be decomposed into three matrices U, V и Σ. The Σ matrix can contain numerous elements, but the greatest significance rests in one or two of its values, while the others are not observed; they do not have any value and are assigned zero.

On the basis of input matrix A, using SVD, the computer program will cluster the respondents into groups according to logical machine learning (artificial intelligence).

## 3. Results

The impact of oral and dental health on daily activities was measured using the OIDP scale, with scores ranging from 0 to 32. The distribution was not in accord with the natural distribution (Table 1; Figure 1), but rather half of the normal/natural distribution was observed.

A large number of adolescents (73%) did not report almost any problems with their mouth or teeth, which may affect their quality of life in the last six months (Figure 1).

OIDP score was dichotomized into two categories depending on whether the respondents had or did not have any oral or teeth problems affecting their quality of life:(1)Absence of oral impact on daily activities (OIDP score = 0);(2)Presence of oral impact on daily activities (OIDP score > 0) (Table 2).

Problems with oral health most frequently had a negative effect on adequate maintenance of oral hygiene (23%), and girls had problems with cleaning teeth more frequently than boys (15%/8%). Oral health also frequently affected enjoying favorite food (10%), as well as sleeping and relaxing (10%). The impact of oral health was the least (4%) on communicating and pronouncing certain words, then on laughing (7%), hanging out with friends (7%), completing school duties (7%) and establishing emotional relationships (8%). Generally, girls had a bigger score than boys on each of the eight items measuring the impact of oral health on quality of life, and therefore the impact of oral and dental health on quality of life was greater for females (Table 2).

In the group of questions related to the OIDP questionnaire on the impact of oral and dental health on daily activities, each of the five possible answers was scored, and each score was assigned an appropriate color (Figure 2):Never or less than once a month—0 points—purple;Once to twice a month—1 point—blue;Once to twice a week—2 points—turquoise;Three to four times a week—3 points—green;Almost every day—4 points—yellow.

When we compare the matrices A and A′, we can observe how much they differ from each other. If the difference is small, the SVD algorithms assign one value, which we marked as (Figure 3)

“True”—yellow color;

If the difference between the two matrices is big, the SVD algorithms assign the other value

“False”—purple color.

As such, artificial intelligence algorithms suggest that we should cluster the respondents into two groups: OIDP = 0 and OIDP > 0.

In the same way, we obtain the second most important matrix A″. This matrix singles out only those respondents who stated the existence of problems in the psychological sphere due to the condition of their mouth and teeth, which is measured by question 5 (“How often in the last six months have you avoided laughing because you were ashamed due to problems with your teeth or mouth?”) (Figure 4):“False”—purple;“True”—yellow.

On the basis of these algorithms, it can be concluded that the mean value of self-efficacy should be lower in respondents with high scores to the question 5 the OIDP questionnaire.

## 4. Discussion

In general, for each of the eight questions, the girls had a more negative attitude than the boys, and therefore the impact of oral and dental health on quality of life was larger female respondents. Similar results have been obtained by a group of our authors, who have reported that the impact of oral health on adolescent quality of life occurred in 53% of adolescent females and 41% of adolescent males [5]. Numerous studies have concluded that women have a worse score as a final result of the analysis of questionnaires dealing with subjective assessment of oral health. The authors have explained this by the level of knowledge and amount of information possessed by women, so their subjective experience of oral health is worse than that of men [18,19,20]. Studies have found that a greater proportion of participants who had never visited a dentist reported less impact. The possible explanation for this fact may be associated with the pattern of symptomatic dental care, since most adolescents visit the dentist only when they have toothache, have a mouth problem, or their own oral health is poor and, statistically, these adolescents are more likely to have more regular dental care, according to the results of some studies [21,22].

Oral health problems most frequently had a negative impact on adequate maintenance of oral hygiene (23%), and girls more often than the boys had problems with cleaning their teeth (15%/8%). Similar results have also been reported by other researchers who have dealt with this matter—24%, 16%, 20% [23,24,25]. With regard to maintaining oral health, girls were generally more responsible than the boys. Additionally, they were more interested in outward appearance, which consequently led to the result that there was a greater impact of oral health on quality of life among girls, because they, in contrast to boys, experienced it as a problem. Satisfaction with dental appearance had a positive correlation with quality of life as a function of oral health [26].

Oral health also frequently affected enjoying favorite food (10%) as well as sleeping and relaxing (10%). Similar results, which show the impact of oral health on diet have also been reported by Chinese researchers [27], while Gajic has reported that as many as 27% respondents had difficulties during eating [5]. Chukwumah reported that the results of the OIDP questionnaire indicated that adolescents most frequently had problems when chewing and maintaining oral hygiene, and as many as 14% of respondents stated that their dental status seriously affects their quality of life [28]. Bianco has concluded that more than two-thirds of adolescents (66.8%), aged 11–16 years, had problems in the previous three months, in their daily life, conditioned by oral health. The difficulties were for the most part linked to chewing (30.4%). Pain in the area of the face and jaws was an important indicator of oral health and its significance lay in the fact that it negatively affected quality of life and could indicate the need for population treatment [29].

Almost all adolescents (94.9%) believed that their dental status had a negative impact on quality of life [30]. On the basis of 11 studies reviewed, Habbu et al. came to the conclusion that knowledge and attitudes related to oral health could be changed by obtaining information and learning new skills, which also indirectly affect the quality of life [31]. The effect on OIDP was associated with negligence by parents, physical abuse, and lack of monitoring among young children and adolescents [32].

The negative impact on the quality of life was more present in the social and psychological spheres of OIDP compared to the functional domain [9]. The impact of oral health was the least (4%) on communicating and pronouncing certain words, then on laughing (7%), hanging out with friends (7%), completing school duties (7%) and establishing emotional relationships (8%). The obtained results suggest that oral health did not have a big impact on the quality of life of adolescents under survey. However, there are large differences in certain results and studies and some authors have reported a far bigger impact on daily activities [28,30]. In the midst of a new pandemic, a decrease in the perception of oral health problems by adolescents during the outbreak of the COVID-19 virus in Brazil has been observed [33].

The presence of malocclusion, as well as the age of adolescents, significantly affects the quality of their lives. Further studies among populations of orthodontic patients are desirable [34].

In one study, adolescents with high sugar consumption showed a greater impact of oral health on quality of life [35]. Untreated caries and its immediate consequence, pain of dental origin, are the main causes of the impact of oral health on adolescent quality of life [36]. Thus, improving behavior among dental visits of low socioeconomic groups would have a greater effect on improving oral health, and thus reducing the impact of oral health on quality of life. The present study demonstrated the independent association between behavioral and psychosocial factors in determining the impact of oral health on quality of life. This demonstrates that health promotion actions should be directed not only to specific actions such as tooth brushing and fluoride application but should also include broader actions directed at contextual factors such as where the individual lives and their family structure [37].

Further research in the field of machine learning should be intensified. It would be good if it could be integrated with clinical practice as much as possible as great benefits could be gained from this technology in the future. All medical documentation should be stored digitally, adequately processed, prepared and ready to be subjected to analyses using artificial intelligence algorithms. At the same time, this creates the basis for continued monitoring of patient health and long-term surveillance of the effects of drugs and therapeutic procedures. This type of research also constitutes the basis for identifying health risks which could be monitored using electronic medical records. Results of studies which have dealt with the application of artificial intelligence in clinical medicine and dentistry have demonstrated that this is actually the most sophisticated area in health care, with excellent prospects for success, but further research and advancement are needed in this area, even more intensive than at present. Research must be more extensive so that high-quality process automation can be devised for discovery of new drugs and new therapeutic procedures [38]. Much better distribution and presentation of results would be achieved if there were large databases that are publicly available. Studies should have a larger sample because the obtained results could be presented with much better quality using neural networks, i.e., machine learning.

Human intuition and machine algorithms came to the same conclusion on how the respondents should be divided. As such, method quality and the need to perform analyses of this type in dentistry studies were demonstrated. Using artificial intelligence algorithms, the respondents can be clustered into characteristic groups that allow the discovery of details not possible with the intuitive division of respondents by gender.

Machine learning in dentistry will be of value for all dental practitioners and researchers who wish to learn more about the potential benefits of using machine learning techniques in their work [7].

AI-based applications will streamline care, relieving the dental workforce from laborious routine tasks, increasing health at lower costs for a broader population, and eventually facilitate personalized, predictive, preventive, and participatory dentistry. However, AI solutions have not by large entered routine dental practice, mainly due to (1) limited data availability, accessibility, structure, and comprehensiveness; (2) lack of methodological rigor and standards in their development; (3) and practical questions around the value and usefulness of these solutions, but also ethics and responsibility [39].

Because of their powerful capabilities in terms of data analysis, these virtual algorithms are expected to improve the accuracy and efficacy of dental diagnosis, provide visualized anatomic guidance for treatment, simulate and evaluate prospective results, and project the occurrence and prognosis of oral diseases [40].

In many areas of dentistry, such as orthodontics and maxillofacial surgery, but also periodontics or prosthetics, only a correct diagnosis ensures the correct treatment plan, which is the only way to restore a patient’s health. The diagnosis and treatment plan is based on specialist knowledge, but is subject to a large, multifactorial risk of error. Therefore, the introduction of multiparametric pattern recognition methods (statistics, machine learning and artificial intelligence (AI)) is of great hope for both physicians and patients [41].

## 5. Conclusions

Comparing artificial intelligence to standard statistical methods would not be a fair comparison. One method is not better than the another, as they are two different ways of analyzing the same data. The future of science lies precisely in machine learning. Dental education will need to accompany the introduction of clinical AI solutions by fostering digital literacy in the future dental workforce.

## Figures and Tables

**Figure 1 children-08-01156-f001:**
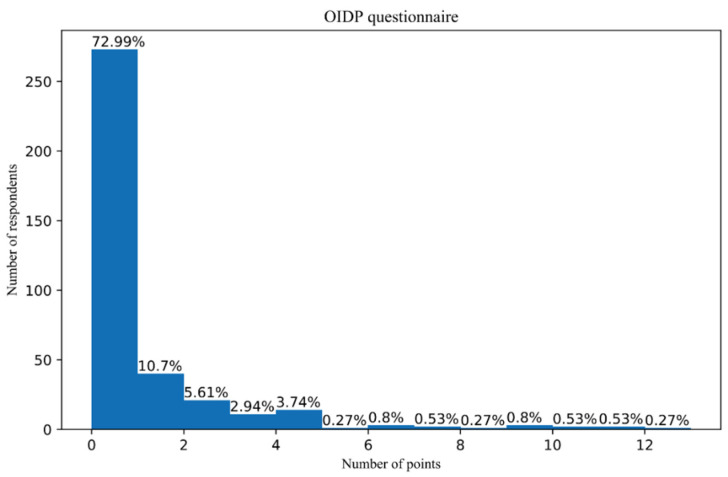
Distribution of the answers to questions from the OIDP questionnaire.

**Figure 2 children-08-01156-f002:**
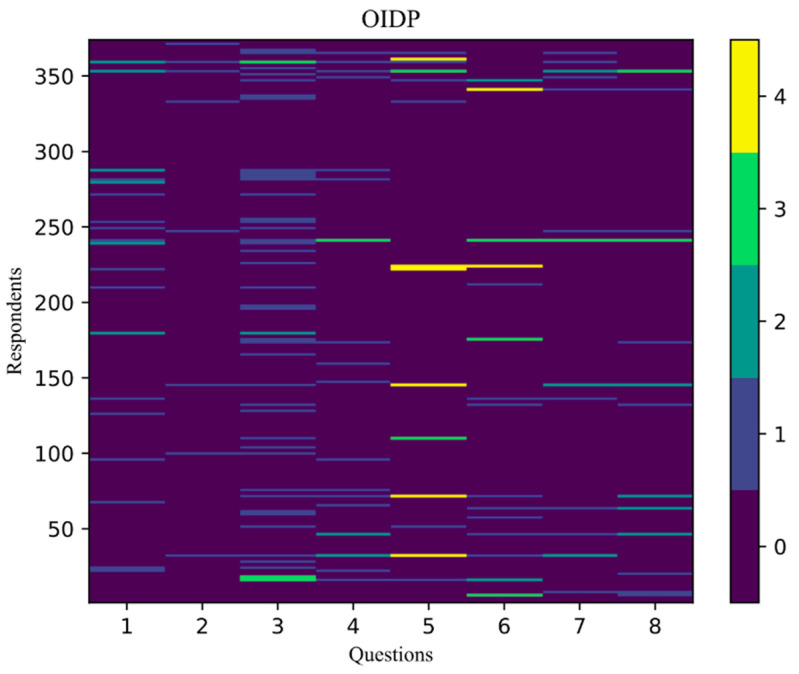
OIDP—initial matrix.

**Figure 3 children-08-01156-f003:**
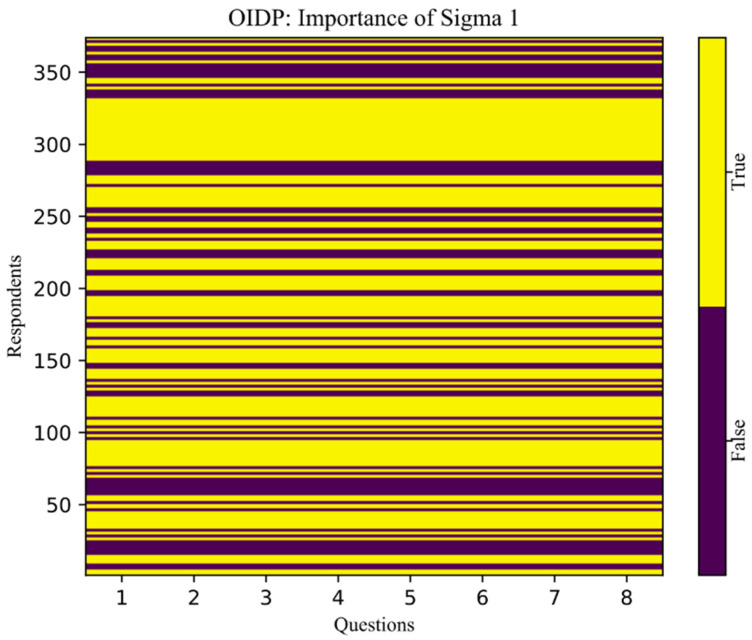
OIDP—Sigma importance 1.

**Figure 4 children-08-01156-f004:**
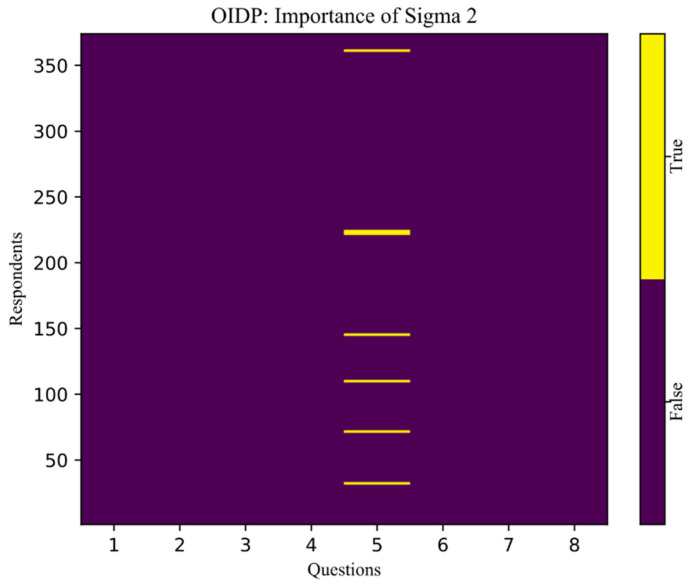
OIDP—Sigma importance 2.

**Table 1 children-08-01156-t001:** Basic statistics—OIDP score.

OIDP Questionnaire *t*-test for Males and Females			
Statistical Parameters	Males	Females	Total
n	128	246	374
Minimum value	0	0	0
Maximum value	14	14	14
Average value	1.28	1.14	1.19
SE	0.33	0.22	0.18
SD	2.64	2.40	2.48
CV (%)	205.79	210.88	209.29
95% CI			
Lower limit	0.62	0.70	0.82
Upper limit	1.95	1.57	1.55
Significance of differences in average values			
*t* = 0.372			
*p* = 0.711			

n—the number of respondents/adolescents, SE—standard value error, SD—standard deviation, CV (%)—coefficient of variation in %, 95% CI—interval of confidence (level of significance 95%), *t*-value for Student’s *t*-test, and *p*-value of statistical significance.

**Table 2 children-08-01156-t002:** Impact of oral and dental treatment on quality of life—dichotomized OIDP score.

Dichotomized OIDP Score for Both Sexes			
Impact of Mouth and Teeth on	OIDP > 0		OIDP = 0
	Males	Females	
Eating and enjoying food	4%	6%	90%
Speaking and pronouncing clearly	0%	4%	96%
Cleaning teeth	8%	15%	77%
Sleeping and relaxing	4%	6%	90%
Smiling, laughing and showing teeth without embarrassment	3%	4%	93%
Maintaing usual emotional state without being irritable	3%	5%	92%
Enjoying contact with people or social role	3%	4%	93%
Carrying out major work	3%	4%	93%

## Data Availability

Data sharing does not apply to this article as no datasets were generated during the current study.

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
