# Peer review of "Analysis of the Impact of Oral Health on Adolescent Quality of Life Using Standard Statistical Methods and Artificial Intelligence Algorithms"

_children, 2021, doi:10.3390/children8121156_

Round 1
Reviewer 1 Report
This is an interesting study and the authors have focused on an area of current interest in using artificial intelligence algorithms in this study. I feel the introduction should contain more information on artificial intelligence and their use in Dentistry. Authors need to present this background information. Objectives need to be clearly stated in the study. Methods should contain sufficient information on the ethics approval which isn't provided in the study. The results section needs more elaboration. I suggest pie charts be used for lucid understanding. The discussion centres more around OID. I suggest authors should focus their discussion on the objectives of the manuscript. The discussion should focus on AI and its applications in Dentistry and Medicine. The conclusions should be succinct.
Author Response
Response to Reviewer 1 Comments
Point 1: This is an interesting study and the authors have focused on an area of current interest in using artificial intelligence algorithms in this study. I feel the introduction should contain more information on artificial intelligence and their use in Dentistry. Authors need to present this background information.
Response 1: Artificial intelligence contemporary dentistry, clearly explain its significance for dental imaging, oral diagnosis and treatment, dental designs and dental research. Machine learning is an emerging field of artificial intelligence research and practice in which computer agents are employed to improve perception, cognition, and action based on their ability to “learn”, for example through use of big data techniques. Its application within dentistry is designed to promote personalized and precision patient care, with enhancement of diagnosis and treatment planning.
---Ko CC, Shen D, Wang L. Machine learning in dentistry.DOI:10.1007/978-3-030-71881-7. ISBN: 978-3-030-71880-0.Springer, January 2021.
Point 2: Objectives need to be clearly stated in the study.
Response 2: The aim of the study was to determine the adolescent oral-health impact on the quality of life and to compare the results obtained by using standard statistical methods and applying artificial intelligence algorithms.
Point 3: Methods should contain sufficient information on the ethics approval which isn't provided in the study.
Response 3: Implementation of this study has been approved by the Ethics Committee of the Faculty of Dentistry in Pancevo (No 795/2-2016 from 26.05.2016).
Point 4: The results section needs more elaboration. I suggest pie charts be used for lucid understanding.
Response 4:
Our goal was to present the importance of applying the artificial intelligence method in data processing, so we did not deal with details. We believe that the greatest quality of this work is precisely in showing the importance of applying different methods for dental analysis, on the one hand the standard statistical methods, and on the other hand the novelty in dental work - the methods of artificial intelligence. We hope that this paper will raise awareness and intrigue other scientists to use additional methods for data processing and display, in addition to the standard statistical methods that are mostly used.
Thanks for the suggestion for the pie charts. The charts related to artificial intelligence are square so we decided to make everything uniform.
Point 5: The discussion centres more around OID. I suggest authors should focus their discussion on the objectives of the manuscript. The discussion should focus on AI and its applications in Dentistry and Medicine.
Response 5: Thank you for the suggestion. That's a very good point. We have inserted a couple of sentences into the discussion that are related to machine learning.
Machine algorithms came to the same conclusion how the respondents should be divided as did the human, or more precisely, the human mind, which intuitively made this division. In this way, the method quality and the need to perform analyses of this type in dentistry studies were demonstrated and proven.
The selected examples amply illustrate the opportunities to employ a machine learning approach within dentistry while also serving to highlight the associated challenges. Machine Learning in Dentistry will be of value for all dental practitioners and researchers who wish to learn more about the potential benefits of using machine learning techniques in their work.
Point 6: The conclusions should be succinct.
Response 6: We agree with you but we need to get the researchers to take the plunge and move forward.
Reviewer 2 Report
The article validates the Serbian version of ODIP using standard statistical methods versus AI algorithms. This must be one of initial papers which have used AI to evaluate these findings, though there are many reviews on this topic
- Advances in technology like AI have potential to impact hugely health care. A small example of AI impacting health care has been demonstrated in this study.
- The study has included 1st and 2nd year dental students and their peers. Usually dental students, even in their pre-clinical years have good or optimal oral hygiene methods. So, it may be implied that they have minimal dental problems. So the oral impact on daily performances would be minimal. Can the authors comment on why the study was not done on the general population? The study being done on dental students with a large number of female population is a limitation of this study.
- It is already known that females are more concerned about dental problems compared to males and seek more often. So what is the need finding apart from this already known finding.
- Sample size estimation, effect size and power analysis has not been calculated
- This is an inherent limitation in the study, which needs to be incorporated in the Limitations section.
- Some English corrections are needed. Grammar, as well as spelling corrections are needed. Page 1, line 36 “may” create , page 3, line 109 “compromise”, Table 2: Speaking; have in line 217
- Why is standard error calculated, when standard deviation is presented?
- Page 7; line 198; results section : The entire paragraph needs to be shifted to discussion.
Author Response
Response to Reviewer 2 Comments
Point 1: Advances in technology like AI have potential to impact hugely health care. A small example of AI impacting health care has been demonstrated in this study.
Response 1: Artificial intelligence contemporary dentistry, clearly explain its significance for dental imaging, oral diagnosis and treatment, dental designs and dental research. Machine learning is an emerging field of artificial intelligence research and practice in which computer agents are employed to improve perception, cognition, and action based on their ability to “learn”, for example through use of big data techniques. Its application within dentistry is designed to promote personalized and precision patient care, with enhancement of diagnosis and treatment planning.
---Ko CC, Shen D, Wang L. Machine learning in dentistry.DOI:10.1007/978-3-030-71881-7. ISBN: 978-3-030-71880-0.Springer, January 2021.
Point 2: The study has included 1st and 2nd year dental students and their peers. Usually dental students, even in their pre-clinical years have good or optimal oral hygiene methods. So, it may be implied that they have minimal dental problems. So the oral impact on daily performances would be minimal. Can the authors comment on why the study was not done on the general population? The study being done on dental students with a large number of female population is a limitation of this study.
Response 2: The study included 1st and 2nd year pupils from Belgrade Secondary Dental School. In Serbia, most of these children do not enroll at the Faculty of Dentistry, but qualify as dental nurses and dental technicians. The school system in Serbia is very similar to that in Germany, where the best students attend Gymnasium as a high school, while the rest attend “realschule” and they do not usually continue their education.
The adolescent population is a vulnerable group with regard to preserving oral health. In the period of turbulent emotional, physical and physiological changes there are many risk factors that can endanger the oral health of adolescents. During this period, the influence of parents reduces, while the influence of the environment and peers increases. There are the smallest differences in behavior that can be related to school readiness.
Point 3: It is already known that females are more concerned about dental problems compared to males and seek more often. So what is the need finding apart from this already known finding.
Response 3: Females are more concerned about dental problems compared to males and seek more often. So we expected a greater impact of the state of oral health on the quality of life of adolescents. Which we showed.
Point 4: Sample size estimation, effect size and power analysis has not been calculated
Response 4: These methods of artificial intelligence are a novelty, so it is very ungrateful to comment on the strength of the analysis and the sample size used because it is difficult to find comparable works that would give an accurate estimate of the strength of the analysis. The sample used is the best we could get in the given circumstances and pandemic situation.
Point 5: This is an inherent limitation in the study, which needs to be incorporated in the Limitations section.
Response 5: This is a very good point. We would like to include comments in the Limitations section regarding our Artificial intelligence method if the paper is accepted.
Point 6: Some English corrections are needed. Grammar, as well as spelling corrections are needed. Page 1, line 36 “may” create , page 3, line 109 “compromise”, Table 2: Speaking; have in line 217
Response 6: A poor status of oral health may create problems to a greater or lesser extent while enjoying food, speaking, sleeping, laughing, cleaning teeth, completing school duties, and creating friendly and emotional relationships.
The selection of rank value of the matrix which is used after applying the SVD algorithm to represent the original image matrix is a compromise between achieving the definite desired degree of compression and keeping the acceptable quality of the digital image.
Speaking and pronouncing clearly
Similar results have been obtained by a group of our authors, who have reported that the impact of oral health on the quality of life occurred in 53% of respondents of female gender and 41% of adolescents of male gender.
Point 7: Why is standard error calculated, when standard deviation is presented?
Response 7: We wanted to present all the different statistical outputs to give as much information as possible to our readers. Yes, you are right we could have exclude a standard error, for example, but we decided to use all the information.
Point 8: Page 7; line 198; results section : The entire paragraph needs to be shifted to discussion.
Response 8: We agree with you, it is better to move the next paragraph to the discussion section.
Machine algorithms came to the same conclusion how the respondents should be divided as did the human, or more precisely, the human mind, which intuitively made this division. In this way, the method quality and the need to perform analyses of this type in dentistry studies were demonstrated and proven.
Reviewer 3 Report
Dear authors,
Thank you for submitting your manuscript!
The aim of this study was to compare the standard statistical methods and artificial intelligence algorithms in the results of the impact on quality of life of adolescent oral health. The overall objective is good. The manuscript needs major adjustments.
General Comments:
Thank you for the authors for their efforts in this manuscript!
Please pay attention to some minor spelling and grammar mistakes.
Introduction:
- Add a paragraph in the introduction talking about the use of artificial intelligence in previous research in Dentistry and its applications.
- Remove the paragraph or sentences about “Matrix decomposition”.
Materials and Methods:
- Specify the sample size calculation, random sample selection procedure and response rate in details.
- Please specify the statistical analysis used to analyze the data.
Results:
- Why is it important to compare the results between the genders (Table 1 and Table 2)?
- Any other factors that should be considered in explaining the findings or comparing the adolescent quality of life beside gender?
- Compare the findings about the two ways of analysis in a better way rather than general comparison.
Discussion:
- The discussion mainly focused on the first aim. How about the second aim in more details?
- Again, please discuss other factors that could explain the first aim findings.
Conclusion:
Need to be improved
Thank you!
Author Response
Response to Reviewer 3 Comments
Introduction:
Point 1: Add a paragraph in the introduction talking about the use of artificial intelligence in previous research in Dentistry and its applications.
Response 1: Artificial intelligence contemporary dentistry, clearly explain its significance for dental imaging, oral diagnosis and treatment, dental designs and dental research. Machine learning is an emerging field of artificial intelligence research and practice in which computer agents are employed to improve perception, cognition, and action based on their ability to “learn”, for example through use of big data techniques. Its application within dentistry is designed to promote personalized and precision patient care, with enhancement of diagnosis and treatment planning.
---Ko CC, Shen D, Wang L. Machine learning in dentistry.DOI:10.1007/978-3-030-71881-7. ISBN: 978-3-030-71880-0.Springer, January 2021.
Point 2: Remove the paragraph or sentences about “Matrix decomposition”.
Response 2: We removed sentence about “Matrix decomposition”. We believe that this methodology may be a bit confusing, at first glance, but we believe that the paragraphs related to matrix decomposition should remain because it is, in fact, the essence of our work and the description of the new methodology.
Materials and Methods:
Point 1: Specify the sample size calculation, random sample selection procedure and response rate in details.
Response 1: This is a very good point. We did our best and knew what we could do in accordance with the new pandemic and the covid situation. We would like to include comments in the Limitations section if the paper is accepted.
All students of one school from two generations were included in the study and patients who attended the faculty. Students were not randomized, all were selected. We considered that a sufficient number for the sample because all children of one school from two generations were included.
Point 2: Please specify the statistical analysis used to analyze the data.
Response 2: The t test (significance test) was used within standard statistical methods, while the emphasis was on methods of artificial intelligence (Singular Value Decomposition), which are not statistical methods.
Results:
Point 1: Why is it important to compare the results between the genders (Table 1 and Table 2)?
Response 1: Comparing the results between males and females is becoming a standard method used in most epidemiological studies, so we have used it in our own, as well.
Point 2: Any other factors that should be considered in explaining the findings or comparing the adolescent quality of life beside gender?
Response 2: We have tried, to the best of our knowledge, to include factors that could be related to quality of life. We did not find any strong correlation that is dominant as a gender difference.
Point 3: Compare the findings about the two ways of analysis in a better way rather than general comparison.
Response 3: Comparing artificial intelligence with standard statistical methods would not be fair. The paper does not want to point out that one method is better than another but wants to present two different ways of analyzing the same data. Also, we would like to encourage other authors to start processing data with the help of non-standard statistical methods, because the future of science lies precisely in machine learning.
Discussion:
Point 1: The discussion mainly focused on the first aim. How about the second aim in more details?
Response 1: Thank you for the suggestion. That's a very good point. We have inserted a couple of sentences into the discussion that are related to machine learning.
Machine algorithms came to the same conclusion how the respondents should be divided as did the human, or more precisely, the human mind, which intuitively made this division. In this way, the method quality and the need to perform analyses of this type in dentistry studies were demonstrated and proven.
Machine Learning in Dentistry will be of value for all dental practitioners and researchers who wish to learn more about the potential benefits of using machine learning techniques in their work.
----Ko CC, Shen D, Wang L. Machine learning in dentistry. DOI:10.1007/978-3-030-71881-7. ISBN: 978-3-030-71880-0. Springer, January 2021.
AI-based applications will streamline care, relieving the dental workforce from laborious routine tasks, increasing health at lower costs for a broader population, and eventually facilitate personalized, predictive, preventive, and participatory dentistry. However, AI solutions have not by large entered routine dental practice, mainly due to 1) limited data availability, accessibility, structure, and comprehensiveness, 2) lacking methodological rigor and standards in their development, 3) and practical questions around the value and usefulness of these solutions, but also ethics and responsibility.
---Schwendicke F, Samek W, Krois J. Artificial Intelligence in Dentistry: Chances and Challenges. J Dent Res 2020 Jul;99(7):769-774.
Because of their powerful capabilities in data analysis, these virtual algorithms are expected to improve the accuracy and efficacy of dental diagnosis, provide visualized anatomic guidance for treatment, simulate and evaluate prospective results, and project the occurrence and prognosis of oral diseases.
---Shan T, Tay FR, Gu L.J Dent Res. 2021 Mar;100(3):232-244. doi: 10.1177/0022034520969115. Epub 2020 Oct 29.PMID: 33118431
In many areas of dentistry, such as orthodontics and maxillofacial surgery, but also periodontics or prosthetics, only a correct diagnosis ensures the correct treatment plan, which is the only way to restore the patient's health. The diagnosis and treatment plan is based on the specialist's knowledge, but is subject to a large, multi-factorial risk of error. Therefore, the introduction of multiparametric pattern recognition methods (statistics, machine learning and artificial intelligence (AI)) is a great hope for both the physicians and the patients.
---Machoy ME, Szyszka-Sommerfeld L, Vegh A, Gedrange T, Woźniak K. The ways of using machine learning in dentistry. Adv Clin Exp Med. 2020 Mar;29(3):375-384.
Point 2: Again, please discuss other factors that could explain the first aim findings.
Response 2:
The presence of malocclusion, as well as the age of adolescents, significantly affects the quality of their lives. Further studies among populations of orthodontic patients are desirable.
---Kolawole KA, Ayodele-Oja MM. American Journal of Orthodontics and Dentofacial Orthopedics.Vol 159, Issue 2, February 2021, Pages e149-e156
In the midst of a new pandemic, a decrease in the perception of oral health problems by adolescents during the outbreak of Covid-19 virus in Brazil has been observed.
---Knorst JK, Brondani B, Tomazoni F, Vargas AW, Cósta MD et all. COVID-19 pandemic reduces the negative perception of oral health-related quality of life in adolescents. Quality of Life Research volume 30, pages1685–1691 (2021).
Studies have found that a greater proportion of participants who had never visited a dentist reported less impact. The possible explanation for this fact may be associated with the pattern of symptomatic dental care, since most adolescents visit the dentist only when they have ttothache, have a mouth problem, or their oral health is poor and statistically, these are more likely to have more regular dental care, according to the results of some studies.
---Pentapati KC, Acharya S, Bhat M, Rao SVK, Singh S. Oral health impact, dental caries, and oral health behaviors among the National Cadets Corps in South India. Journal of investigative and clinical dentistry. 2013;4:39-43.
In one study, adolescents with high sugar consumption showed a greater impact on quality of life in relation to oral health. Untreated caries and its immediate consequence, pain of dental origin, are the main causes of the impact on the quality of life of adolescents. Thus, improving behavior among dental visits of low socioeconomic groups would have a greater effect on improving oral health, reducing the impact on quality of life related to oral health. The present study demonstrated the independent association between behavioral and psychosocial factors in determining the impact on the quality of life related to oral health. This demonstrates that health promotion actions should be directed not only to specific actions such as tooth brushing and fluoride application but should include broader actions directed at contextual factors where the individual lives and their family structure.
---Veras S, Kozmhinsky V, Goes P, Heimer M. Behavioral and Psychosocial Factors as Mediators of the Oral Health Impact on Adolescents Quality of Life. January 27th 2020 DOI: 10.5772/intechopen.89567
---Keles S, Abigail F, Adana F. Oral health status and oral health related quality of life in adolescent workers. Clujul Medical. 2018;91:462-468
---Crocombe LA, Broadbent JM, Thomson WM, Brennan DS, Poulton R. Impact of dental visiting trajectory patterns on clinical oral health and oral health-related quality of life. Journal of Public Health Dentistry. 2012;72(1):36-44.
Conclusion:
Point 1: Need to be improved
Comparing artificial intelligence with standard statistical methods would not be fair. One method is not better than another, they are two different ways of analyzing the same data. The future of science lies precisely in machine learning. Dental education will need to accompany the introduction of clinical AI solutions by fostering digital literacy in the future dental workforce.
Round 2
Reviewer 1 Report
Thanks for addressing the comments.
Reviewer 3 Report
The authors did well in improving their manuscript.
Thank you for their efforts!